# OpenReview forum: "Automated Hypothesis Validation with Agentic Sequential Falsifications"
_ICML.cc/2025/Conference — ICML 2025 poster_

### Official Review · Reviewer_yhmP · 2025-03-10

**Overall Recommendation:** 4

**Summary:**

This paper presents POPPER, an agent framework inspired by Karl Popper's principle of falsification that can automatedly validate hypotheses statistically rigorously. POPPER validate a hypothesis through conducting analyses or experiments for each sub-hypothesis, calculating p-values and e-values, and determining whether to reject the global null hypothesis. Experiments were conducted on two benchmarks: TargetVal that addresses geneotype-phenotype hypotheses in biology and DiscoveryBench that spanning six domains including sociology, biology, humanities, economics, engineering, and meta-science. Results show that the proposed POPPER succeeds in controlling Type-I error under 0.1 across all datasets and also achieves significant power improvement against various baselines. Also, human study show that POPPER and human perform equally well on selected tasks, with POPPER being more efficient for spending less time and conducting more statistical tests.

## update after rebuttal
Thank you for your response! The observations on success mode are within expectation. Segmenting the hypothesis into a sequence of more verifiable sub-hypotheses is helpful, with rigorous statistical test providing suffcient evidences to validate. It is good to add some rigorous experiments on how these two elements respectively affect the result, just as POPPER-NoReleCheck presenting the result of removing relevance checker. I decided to maintain my original score.

**Claims And Evidence:**

Yes, the claims made in the submission are supported by clear and convincing evidence.

**Essential References Not Discussed:**

This paper proposes to automatedly validate hypotheses via Popper's principle of falsification. However, similar idea of utilizing falsification for scientific discovery was first discussed in [1], which should be an essential reference but not mentioned in this paper.

[1] Liu et al., 2024. AIGS: Generating Science from AI-Powered Automated Falsification. arXiv:2411.11910.

**Experimental Designs Or Analyses:**

Yes, I checked the soundness of the experimental designs and analyses.

**Methods And Evaluation Criteria:**

Yes, the proposed methods and evaluation criteria make sense for the research problem.

**Other Comments Or Suggestions:**

1. Typo: several left double quotation marks are incorrectly written as right double quotation marks in both text and tables.
2. Nitpick: the use of * to indicate the best results in Tabel 3 and Table 4 lead to the misalignment of figures vertically. (It is my personal preference that figures should better be aligned and this nitpick can be simply neglected.)

**Other Strengths And Weaknesses:**

**Strengths**

1. This paper proposes to validate hypotheses automatedly via statistical tests, which is not involved in previous work. Such method successfully ground the process of hypothesis validation on numerical analysis instead of pure language based analysis, which is more rigorous and reliable, and is helpful to reduce hallucinations for research tasks. Therefore, this paper may greatly advance the paradigm of research agent and I believe it can be a great contribution to the community.

**Weaknesses**

1. This paper conducts analysis on potential failure modes of the proposed POPPER. However, human study on the pattern how POPPER outperforms the baselines is not presented. The absence of such analysis may cause the incomplete understanding of why and how POPPER actually works.
2. The proposed statistical test based analysis is powerful, but however, it is questionable whether it can be applied to more research areas and hypotheses. For example, for mathematical theory proof tasks or linguistic research, statistical analysis may not be applicable.

**Questions For Authors:**

1. Analysis for the action mode of POPPER is now only conducted on failures instead of successes. Can authors conduct analysis on how POPPER outperforms the baselines? Is it because that it can propose better experiments or the explicit calculation of p-values and e-values is the most important factor? Such analysis can be helpful to better understand the mechanism of POPPER.
2. As described in the paper, the design agent is presented with the details of how to conduct the experiment in a given domain. Can authors elaborate to what extent is such assistance provided to the agent, i.e. only a brief description of how the experiment might be conducted / explanation of how to conduct the experiment step by step but without the coding details / all detailed code or function needed for the experiment. Furthermore, are all the possible experiments or only a subset of them are provided as assistance? Do agents strictly follow the given instructions or can they come up with some experiments that are not presented in the assistance?

**Relation To Broader Scientific Literature:**

This paper proposes to validate hypotheses via statistical tests, while previous work mainly utilize natural language directly for hypothesis validation. Such method leads to more rigorous validation and can effectively reduce hallucinations generated by LLMs, which is a solid contribution of this paper.

**Theoretical Claims:**

Yes, I checked the correctness of the proofs for theoretical claims and there are no issues.

---

> ### Author Rebuttal · Authors · 2025-03-28
>
> We thank the reviewer for their positive feedback! We respond to the specific comments below:
>
> > **“Can authors conduct analysis on how POPPER outperforms the baselines?”**
>
> We appreciate the reviewer’s insightful suggestion. Following the reviewer’s recommendation, we conducted a detailed manual examination and identified three main reasons for POPPER's superior performance:
>
> 1. The sequential experiments employed by POPPER significantly improves power over methods like CodeGen, ReAct, and Self-Refine, which uses only 1-2 experiments. In particular, many of the hypotheses cannot be directly observed, and therefore a direct attempt at validating the hypothesis might often fail, whereas Popper allows a sequence of carefully designed implication tests that can find more meaningful results.
>
> 2. Self-refine and relevance checker also help refine the experiment design, whereas the CodeGen, ReAct, and Self-Refine uses the most obvious experiment, which could contain bias and lacks rigor.
>
> 3. With its rigorous e-value-based approach, POPPER addresses the sequential dependencies of the falsification experiments and safely aggregates p-values from multiple experiments. This enables POPPER to achieve better Type I error control compared to the Fisher combined test and the LLM-likelihood. The Fisher combined test is not well-calibrated, and LLM-estimated scores often exhibit bias.
>
> We will add these insights to our revised manuscript.
>
> > **“The design agent is presented with the details of how to conduct the experiment in a given domain. Can authors elaborate on the extent of such assistance provided to the agent?”**
>
> Thank you for this important clarification request. To clarify, we did not provide explicit experimental details or instructions to the agent for each domain. The only domain-specific information provided was: “You are an expert specialized in the field of {domain},” as outlined in Supplementary Notes Listing 2. Consequently, the LLM agent designs experiments solely based on its internal world knowledge without additional domain-specific guidance.

---

> > ### Comment · Reviewer_yhmP · 2025-04-03
> >
> > Thank you for your response! The observations on success mode are within expectation. Segmenting the hypothesis into a sequence of more verifiable sub-hypotheses is helpful, with rigorous statistical test providing suffcient evidences to validate. It is good to add some rigorous experiments on how these two elements respectively affect the result, just as POPPER-NoReleCheck presenting the result of removing relevance checker. I decided to maintain my original score.

---

### Official Review · Reviewer_Ah4v · 2025-03-12

**Overall Recommendation:** 4

**Summary:**

The paper introduces POPPER, a framework for using AI agents to perform hypothesis validation. Given a hypothesis, the system designs and executes a series of falsification experiments, and uses statistical methods for accumulating evidence until the hypothesis can be accepted or rejected with a final p-value.

Careful statistical control of Type-I errors and power analysis allows balanced understanding of the system's reliability, and the authors find that an instantiation of POPPER using GPT-4o significantly outperforms baselines on a set of static datasets as well as interactive simulations.

In a human expert study, their instantiation also matches the power and Type-I error rates of computational biologists and bioinformaticians on hypothesis testing on biology datasets.

**Claims And Evidence:**

The core results of Section 4.1 appear supported by their results for their chosen settings.

The claim "POPPER compares with human experts" should be appropriately caveated with limitations given the small sample size of human experts (only 9 experts compared) and limited complexity of the settings tested (the static datasets with clean variable headers are likely easier than situations one might encounter in reality).

**Essential References Not Discussed:**

In "LLM for hypothesis testing and experiments" (appendix), the authors may consider including literature on LLMs for automated ML experimentation, such as RE-Bench (Wijk et al) and MLE-bench (Chan et al), which pose open-ended problems for LLM agents to attempt making progress on.

**Ethical Review Concerns:**

N/A.

**Experimental Designs Or Analyses:**

- The experimental setup of Section 3 makes sense to me, i.e. the choice of static datasets as the environment for ease of implementation. In future work, I would be excited to see POPPER attempt something like DiscoveryWorld (Jansen et al) which gives a different flavor of experimentation than static data analysis.

**Methods And Evaluation Criteria:**

- POPPER as a framework is nicely designed and feels like a flexible framework for hypothesis validation.
- The use of Type-I error and Power as success metrics are sensible and useful.

**Other Comments Or Suggestions:**

None.

**Other Strengths And Weaknesses:**

Strengths
- The POPPER framework is very clearly presented and has an elegant form, which appears to broadly support any sort of hypothesis testing. I like the sequential application of "Experiment Design" and "Experiment Execution" on sub-hypotheses to yield evidence that can be accumulated to evaluate the main hypothesis.
- Impressive results in Table 3 as well as on the human experts study.
- I liked the additional analyses and ablations the authors provided: the comparison between different LLMs, ablation results of NoReleCheck (and human annotations), error analysis, and especially the comparison to human baselines was an important and useful addition.

Weaknesses
- When reading the paper, I was looking for a section on Limitations. I found this in the appendix as part of the supplementary material (thank you for including this!), but I think it is important to mention these limitations in the main paper, even if just deferring readers to the appendix. As it stands, the main paper sounds like POPPER can be treated as a "solve all" method for automating science, but as I understand from the appendix there are important caveats to be mindful of, e.g. "Type-I error v.s. false discoveries". Acknowledging this in the main text would help readers have a more measured understanding of the work.

**Questions For Authors:**

None. Thank you for your paper!

**Relation To Broader Scientific Literature:**

This paper fills a useful role in tying together work in hypothesis generation and experiment execution (already discussed in the appendix "Full related works" section), which are typically studied separately and lack a statistically-rigorous framework to tie everything together. A structured framework like POPPER will grow in usefulness as both hypothesis generation methods and experiment execution methods (as well as the underlying models) improve.

**Theoretical Claims:**

I did not get to check the theoretical claims or proofs, but these would be important for complete assessment of the paper.

---

> ### Author Rebuttal · Authors · 2025-03-28
>
> We thank the reviewer for their positive feedback! We address each point in detail below:
>
> > **“The claim "POPPER compares with human experts" should be appropriately caveated with limitations.”**
>
> We appreciate the reviewer highlighting this important point. We agree and will explicitly note the caveats related to the small sample size and differences from real-world complex settings in the revised manuscript.
>
> > **“The authors may consider including literature on LLMs for automated ML experimentation, such as RE-Bench (Wijk et al.) and MLE-bench (Chan et al.).”**
>
> We thank the reviewer for suggesting additional relevant references. We will incorporate these references in our revised version.
>
> > **“The main paper sounds like POPPER can be treated as a "solve all" method for automating science, but as I understand from the appendix, there are important caveats to be mindful of.”**
>
> We thank the reviewer for this valuable feedback. We agree that clearly presenting the limitations is crucial to ensure responsible usage and prevent potential misuse. We will add detailed discussions on failure modes and Type I error considerations in the main text in the revised manuscript.

---

### Official Review · Reviewer_hGJ8 · 2025-03-14

**Overall Recommendation:** 1

**Summary:**

The paper introduces a framework called Popper, which leverages Large Language Models to validate hypotheses specified in natural language. The proposed framework makes use of two LLM agents; one decomposes the hypothesis of interest into smaller sub-hypotheses and proposes experiments to test them, and another that executes the experiments designed by the first agent. The results of these experiment executions are combined through the use of e-values, derived from p-values produced by the analysis performed by the second LLM agent. A theorem is provided that justifies that the method used to combine the e-values results in a controlled Type-I error for the whole system. The framework is experimentally validated using some recently proposed benchmarks spanning six domains, where it is shown to be the only effective approach. Moreover, a comparison is provided with (human) bioinformaticians, with the proposed framework behaving similarly to the humans while completing the analysis an order of magnitude faster.

**Claims And Evidence:**

* The paper claims to investigate a novel problem setting---validating hypotheses specified in natural language. They discuss prior literature that investigates the most similar problem settings. As far as I am aware, this claim is true.
* It is claimed that the proposed framework is able to design and execute any type of experiments, including laboratory procedures, simulations, or data analyses. This claim is not justified. There is no evidence provided that demonstrate the proposed agents are able to design experiments that can be carried out in a laboratory setting. Moreover, not enough details are provided to determine whether the evaluation includes simulation-based experiments. My understanding of the experimental evaluation is that only data analyses are included, but I am not certain that simulations are excluded.
* The manuscript claims that the Popper framework is able to maintain statistical rigor through the use of a novel sequential testing framework that aggregates evidence from tests. The evidence that this part of the framework is correct comes in the form of Theorem 4, which seems to be true.
* It is claimed that Popper achieves Type-I error control, substantiated by empirical evidence gathered using the DiscoveryBench and TargetVal-IL2 datasets. While I think the experiment referenced here is interesting, I think this claim needs to be toned down. Control of Type-I errors is usually established formally; one has the guarantees that, as long as the assumptions of the test are satisfied, the Type-I error is controlled. This is not the case here.
* It is claimed that Popper has significant power improvements, substantiated by empirical evidence gathered using the DiscoveryBench and TargetVal-IL2 datasets. For similar reasons as the Type-II error claim, I think this one needs to be toned down slightly; the power is determined either analytically or through, e.g., Monte Carlo simulations that will converge towards the true power value. Moreover, I disagree with the exclusion of most methods from this comparison on the basis of poor Type-I error control.
* Popper is demonstrated to be comparable with human experts. This claim is supported by a user study based on an empirical comparison with nine bioinformaticians. I think the claim, as it pertains to power and Type-I error comparisons, is relatively well supported. The comments about efficiency gains should be slightly more nuanced; the improved in time is positive, but I think one could justifiably interpret requiring more code and hypothesis tests as a decrease in efficiency.

**Essential References Not Discussed:**

I have no concerns in this area.

**Ethical Review Concerns:**

There are two ethical issues with the paper:

1) The authors claim to develop a novel statistical testing framework, but this framework already exists. Key papers involved in the development of this framework were cited in the submission without acknowledging that these papers had already developed the proposed statistical framework. Moreover, when confronted with this, the authors now lie that the original manuscript acknowledges the prior work developed the framework.

2) The paper contains a human study, but there is no discussion of obtaining ethics approval.

**Ethical Review Flag:**

Flag this paper for an ethics review.

**Ethics Expertise Needed:**

["Research Integrity Issues (e.g., plagiarism)", "Responsible Research Practice (e.g., IRB, documentation, research ethics, participant consent)"]

**Experimental Designs Or Analyses:**

I think the design of the experiments for determining the Type-I error and Power of the methods is reasonably sound, and I think they allow for making interesting claims. However, I would have appreciated more discussion about the limitations of the conclusions given the source of the data. In particular, the extent to which the hypotheses included in the evaluation are already present in the pre-training data of the the LLMs is unclear. As such, it is not obvious how well the proposed framework will generalise to novel discovery problems.

The comparison with human experts, which based on a small sample size, is still interesting. There are quite limited details provided in the main paper about how this study was carried out but, as with other experimental results, the uncertainties in the estimates are clearly quantified.

**Methods And Evaluation Criteria:**

The method is only presented at a high level of detail; it appears to consist of a pipeline of prompts coupled with standard tools from the e-values literature. It is not clear how the method (i.e., prompts) were constructed. The choice of benchmark datasets is sensible.

**Other Comments Or Suggestions:**

I think the experimental analysis could put a lot more emphasis on determining how robust the framework is. In particular, undertaking intrinsic analyses of the individual components of the system to determine where failures are introduced would be valuable.

**Other Strengths And Weaknesses:**

A major strength of this paper is the first demonstration of a framework that can tackle the problem of falsifying realistic natural language hypotheses.

The major weakness of this paper is the substantial amount of overclaiming. This cannot be overlooked just because of the substantial strength mentioned previously. The way the paper is currently written has the potential to be very misleading, and the claims should be toned down and appropriately caveated. For example, I think the level of emphasis on rigor and statistical guarantees of the proposed pipeline would likely lead some readers to believe that the LLM components of the pipeline are guaranteed to correctly identify and implement the appropriate hypothesis tests for each sub-hypothesis. It should be made much more clear that no such guarantees are provided.

**Questions For Authors:**

Is there a distinction between hypothesis validation and hypothesis testing in the context of this work?

What are the two Power lines in the plot in Fig 4(2)?

**Relation To Broader Scientific Literature:**

This submission addresses a novel formulation of the problem of using machine learning for scientific discovery. Rather than suggesting hypotheses, or focusing on executing experiments, the emphasis is on building a complete framework that can take a hypothesis and falsify it.

**Theoretical Claims:**

The theorem seems to be correct, but I am concerned about novelty here. From what I can tell this is a restating of a fairly central result in the e-values literature. See, e.g., Ramdas et al. (2023), who provide an overview of this area and present essentially the same reasoning for why e-values can be used in this way.

Ramdas et al. "Game-Theoretic Statistics and Safe Anytime-Valid Inferece". In Statistical Science, 2023. https://doi.org/10.1214/23-STS894

---

> ### Author Rebuttal · Authors · 2025-03-28
>
> We greatly appreciate the reviewer's thoughtful feedback and acknowledgment of our work's value in falsifying natural-language hypotheses. Below, we respond in detail to the specific points raised:
>
> > **"Proposed framework claims to design and execute any type of experiments (laboratory, simulations, data analyses)"**
>
> We appreciate the reviewer highlighting this point. Our theoretical framework is indeed valid across various experiment types. Since for wet-lab experiments, the data is collected on the fly, so it naturally satisfy the assumption stated in Section 2. We thus emphasize the broad scope of our framework. However, due to practical constraints (e.g., cost and time), we instantiated our approach specifically through data analysis experiments in our large-scale evaluation. We will revise the text to explicitly acknowledge this practical limitation.
>
> > **"Power determined analytically or via Monte Carlo simulations that converge toward true power values"**
>
> Thank you for raising this point. Currently, our power analysis is conducted using Monte Carlo simulations with five random seeds, and we reported the mean and standard deviations in our result tables. While we acknowledge that additional runs could further reduce variance and improve convergence toward the true power, scaling up significantly is not feasible due to computational constraints.
>
> > **"Disagreement with exclusion of methods lacking proper Type I error control"**
>
> We respectfully clarify that since methods lacking proper Type I error control can inflate false positives, including them in power analysis would be an unfair comparison. We also stress the importance of valid error control as the first-order criterion. We welcome additional elaboration from the reviewer on the disagreement and are eager to discuss this further.
>
> > **"Concern about novelty regarding the theorem presented"**
>
> As explicitly stated in our original manuscript, “Theorem 4 is a standard result following Grunwald et al. (2020), included in Appendix A.2 for completeness.” Our novelty claim does not rest on the safe testing framework itself but on leveraging this framework to instantiate a sequential falsification framework to enable practical and rigorous validation of abstract, free-form hypotheses in LLM-driven experiments. We will clarify this distinction explicitly in the revision.
>
> > **"Extent to which evaluated hypotheses might be present in LLM pre-training data is unclear"**
>
> We appreciate this insightful comment. Our validation approach strictly relies on aggregated e-values derived from statistical analyses grounded in data. In our experiments, as detailed in section 4, we used data permutations for controlling Type I error, which ensures the resulting data is independent of the pre-training data. This experiment setup enforces that any discovery must be purely data-driven and not reliant on the agent's prior knowledge. We will clarify this critical detail in the revision.
>
> > **"Major weakness is substantial overclaiming"**
>
> We thank the reviewer for this important feedback. Although we initially aimed to mitigate overclaiming by carefully delineating assumptions and providing comprehensive failure-mode analyses, we recognize the need for further clarity. In the revision, we will rigorously address ambiguous claims and explicitly highlight caveats to avoid potential misunderstandings. In particular, we will emphasize that the error control relies heavily on our assumptions, and several design components, such as relevance checker and using LLMs with strong capabilities, are intended to make sure the assumptions are satisfied. In practice, users should be careful in judging whether their system is sufficiently powerful to obey these assumptions.
>
> > **"Intrinsic analyses of individual components to determine sources of failure would be valuable"**
>
> Thank you for raising this important point. In section 4.2 and Supplementary Section G, we conducted human annotation on the quality of falsification experiment subhypotheses and the relevance checker’s performance. In Supplementary Section D, we conducted an extensive intrinsic failure-mode analysis by examining detailed logs from 128 failure cases, categorized into 10 distinct failure modes. Additionally, we performed trajectory analysis documented in Supplementary Section E. We will ensure this comprehensive analysis is more prominently highlighted in the manuscript.
>
> > **"Distinction between hypothesis validation and hypothesis testing?"**
>
> We appreciate this clarifying question. We used the terms interchangeably, recognizing that "hypothesis validation" tends to resonate more within scientific domains, whereas "hypothesis testing" is predominantly used in statistical contexts.
>
> > **"Clarify the two power lines in Figure 4(2)"**
>
> The upper line represents statistical power, and the lower line represents the Type I error rate, both plotted against the number of maximum tests conducted.

---

> > ### Comment · Reviewer_hGJ8 · 2025-04-03
> >
> > >Our theoretical framework is indeed valid across various experiment types.
> >
> > It is incorrect to claim that the proposed framework is valid in all of those settings without actually validating the framework in those settings. This is substantial overclaiming and needs to be changed.
> >
> > > We also stress the importance of valid error control as the first-order criterion.
> >
> > No justification is given in the paper or rebuttal stating why Type I errors are more important than Type II errors. This is likely very context dependent. If we instead decide that Type II is more important, the conclusions of the paper change completely.
> >
> > > As explicitly stated in our original manuscript, “Theorem 4 is a standard result following Grunwald et al. (2020), included in Appendix A.2 for completeness.”
> >
> > This text is not in the manuscript. In fact, the text before the statement of Theorem 4 cites Grunwald et al. (2020) only to establish a technical condition. There is no mention that the Theorem is already known.
> >
> > > Our validation approach strictly relies on aggregated e-values derived from statistical analyses grounded in data.
> >
> > The LLM agent is given the freedom to decide how this analysis is performed. If the experiments are based on known phenomena, for which we have already conducted successful analyses that appear in the training data, there is no guarantee that the proposed framework will generalise to new problems. This should be discussed.

---

> > > ### Author Response · Authors · 2025-04-03
> > >
> > > Thank you for your thoughtful follow-up! We recognize that our initial rebuttal did not clearly convey the scope and limitations of our claims, which may have led to misunderstandings. Below, we address each point in more detail to clarify our position.
> > >
> > > > “It is incorrect to claim that the proposed framework is valid in all of those settings without actually validating the framework in those settings. This is substantial overclaiming and needs to be changed.”
> > >
> > > We appreciate the reviewer’s attention to this important issue. We agree and acknowledge that our original phrasing was overly broad, and we will revise the manuscript to make our claims more precise.
> > >
> > > Our intention was to convey that the framework is **theoretically** valid in a broader set of settings, provided that **all three key assumptions are satisfied**. The crucial remaining step—consistent with the reviewer’s observation—is to design a system that meets those assumptions and empirically demonstrates that the error control properties hold in practice. However, we recognize that our previous language may have blurred the line between theoretical potential and empirical evidence. We highlighted that wet-lab experiments may naturally satisfy Assumption 2 (new data collection being independent or conditionally independent of prior evidence), whereas this is a challenge in static data analysis and requires careful LLM design. Since we have not validated the framework in wet-lab settings, we will revise the manuscript to explicitly restrict our claims to static data analysis, where empirical validation has been performed, and significantly tone down any broader claims.
> > >
> > > > “No justification is given in the paper or rebuttal stating why Type I errors are more important than Type II errors. This is likely very context dependent. If we instead decide that Type II is more important, the conclusions of the paper change completely.”
> > >
> > > We thank the reviewer for this thoughtful comment. We fully agree that the relative importance of Type I versus Type II errors is context-dependent. Our intention was not to argue that Type I error is universally more important. In contrast, our intention was that **if a method fails to control Type I error, then apparent improvements in power (i.e., lower Type II error) can be biased**. For example, a method that accepts all hypotheses would show maximal power, yet its conclusions would be invalid due to uncontrolled false positives. Our emphasis on Type I error control is therefore to ensure fair and interpretable comparisons of Type II error performance. We will revise the manuscript to clearly articulate this rationale and avoid any implication that Type I error is inherently more important.
> > >
> > > > “This text is not in the manuscript. In fact, the text before the statement of Theorem 4 cites Grunwald et al. (2020) only to establish a technical condition. There is no mention that the Theorem is already known.”
> > >
> > > We sincerely apologize for the oversight. The discrepancy stems from referencing an updated internal draft that includes citations and clarifications not present in the submitted version. We will ensure the revised manuscript properly acknowledges Grunwald et al. (2020) and clearly states the novelty and context of Theorem 4. We’re grateful to the reviewer for highlighting this.
> > >
> > > > “The LLM agent is given the freedom to decide how this analysis is performed. If the experiments are based on known phenomena, for which we have already conducted successful analyses that appear in the training data, there is no guarantee that the proposed framework will generalise to new problems. This should be discussed.”
> > >
> > > Thank you for this insightful observation. We apologize for any confusion caused by our previous rebuttal.
> > >
> > > We completely agree that the overlap between the training data and the experimental tasks poses a potential data leakage risk. In our earlier response, we intended to convey that **in our experiments, the evaluation setup of Type I error under the null is justified**. Specifically, we evaluated Type I error using permuted data, which ensures that the data is under the null, regardless of whether the original (unpermuted) data is present in the LLM’s training set, thus providing a way to isolate the framework’s behavior from potential data overlap. That said, we fully acknowledge that potential data leakage may impact our power estimation—a concern broadly relevant to any tasks using public datasets—and thus should be treated with care.
> > >
> > > We will revise the manuscript to explicitly discuss this limitation, describe how we attempted to mitigate it in our experiments, and potential strategies such as using unpublished datasets or probing the likelihood of training data overlap.
> > >
> > > ⸻
> > >
> > > Please let us know if any further clarification would be helpful. We sincerely appreciate the reviewer’s detailed and constructive feedback—the emphasis on rigor is especially valued and will greatly enhance the quality of our work.

---

### Official Review · Reviewer_aYxm · 2025-03-18

**Overall Recommendation:** 3

**Summary:**

Manuscript provides a contribution to the automated scientist literature. Premise is that free-form hypothesis positing and testing needs to be accomplished at scale and this necessitates automation. This task is accomplished using agentic/LLM flows which break-down a hypothesis into sub-hypotheses. Sub-hypotheses are sequentially tested and resulting e-values combined using a rigorous procedure which controls for Type I error. Proposal of sub-hypotheses from a free-form hypothesis is achieved via a sequence of LLM prompts with canonical – chain of thought – approaches to obtaining often-valid reasoning chains. Despite shortcomings, this approach compares favorably in terms of efficiency with trained experts performing the same task of hypothesis validation, while at the same time making just as few mistakes as data scientists and statisticians.

**Claims And Evidence:**

Chief claim that the manuscript makes is that the method achieves statistical rigor as a result of theoretically sound approach to combining e-values when executing a sequence of sub-hypotheses tests.

Key assumption is that the hallucinations and unintentional introduction of an irrelevant sub-hypotheses would be caught through self-refinement (an LLM procedure) or relevance check (another LLM based procedure). Humans being susceptible to such mistakes as well, authors compare performance of their method POPPER to performance of trained statisticians. However, sentences like:  “By integrating a sequential testing paradigm with automated experiment design and execution, POPPER delivers scalable, statistically rigorous hypothesis validation” extend the claim of statistical rigor obtained under assumption of sensible hypothesis selection are not obviously applicable when an LLM can swap in a sub-hypothesis or report incorrect p-values.

Prompts in the provided code such as: "**IMPORTANT**
You should ALWAYS report a p-value EXACTLY AS IT IS. If a p-value is 4.2e-01, report 4.2e-01, DO NOT REPORT 4.2e-02!" are indicative of challenges in using LLMs to process outputs of numerical methods.

**Essential References Not Discussed:**

N/A

**Experimental Designs Or Analyses:**

Using ChatGPT-o1 to assign failure modes and then sanity check a subset of those assessments seems to reliant on LLMs that need guidance as shown above. Please assess more than 30 examples of hypothesis validation failure modes especially since this is not an expensive effort compared to what has been done already.

It would be very useful to see how susceptible POPPER is on simulated data where the truth is known ahead of time and performance can be assessed without recourse to LLM critics. To recall the instruction to agents not to mess up quantitative data, testing with a range of p-values and hypothesis sequences would reveal any susceptibility of the method to ranges of p-values. At the very least, permute names of the diseases and genes and see whether the findings tend to be sensible or the generated hypotheses ignore the evidence.

**Methods And Evaluation Criteria:**

Yes, assessing construction of sub-hypothesis in biological domains where cause of difference in expression can range across variety of factors: expression of causal genes, genetic variation, cell and tissue specific milleu, post-translational modifications etc. are well suited for assessing whether the proposed sub-hypothesis make biological sense. Comparing to trained experts is a sensible and practical baseline.

**Other Comments Or Suggestions:**

Typo:
“Assumption 2 requires the e-value in each iteration is valid
conditional on prior information.
“
Missing “that” after requires

**Other Strengths And Weaknesses:**

Taking a stab at statistical rigor when using LLMs is bold and laudable. With toned down language and claims, I think this paper can help start the conversation about how inherently noisy LLMs with poorly understood distributions can never-the-less be incorporated into statistically sane procedures.

In the opposite direction, mislabeling the method as a statistically rigorous has potential to devalue the label.

**Questions For Authors:**

How hard would it be to create and run a synthetic experiment with a known ground truth and simulated data of expression or at least mock the p-values and results fed to POPPER? Would permuting gene names across the datasets violate any of the assumptions of the method? For a gentler approach, would it be informative to run POPPER on an alternative reality where genes are subtly renamed (interleukins and other signaling molecules exchanging their names) to see if the IL2 becoming IL9 or CXCL9 would be a bridge too far for POPPER?

**Relation To Broader Scientific Literature:**

This work fits squarely into the automated-scientist literature, but it aims at a higher degree of rigor. Based on my understanding of the paper it does not quite reach the desired levels, but is none the less complementary to the existing work.

**Theoretical Claims:**

I have read proof of theorem 4. No objections.

---

> ### Author Rebuttal · Authors · 2025-03-28
>
> We sincerely thank the reviewer for the constructive feedback and for recognizing our efforts to introduce rigor in the context of LLMs as bold and commendable. We address the thoughtful suggestions raised by the reviewer in detail below:
>
> > **"Tune down the claim and specify the assumptions"**
>
> We appreciate this suggestion and fully agree. We have explicitly highlighted all underlying assumptions in Section 2, discussed Popper's reliance on the base LLM's reasoning capabilities in Section 4, and addressed limitations and failure cases through detailed error analysis in the appendices of our initial submission. We will further clarify ambiguous statements and ensure all claims are appropriately moderated in the revised manuscript. In particular, we will emphasize that the error control relies heavily on our assumptions, and several design components, such as relevance checker and using LLMs with strong capabilities, are intended to make sure the assumptions are satisfied. In practice, users should be careful in judging whether their system is sufficiently powerful to obey these assumptions.
>
>
> > **Prompts in the provided code are indicative of challenges in using LLMs to process outputs of numerical methods.**
>
> We appreciate the detailed feedback. We acknowledge that p-hacking and misreporting p-values were indeed an issue in the initial experiments, especially when the base model is weaker. However, with the additional self-refine and other prompting mechanisms, we were able to consistently control the type I error rate with Claude 3.5 Sonnet. We discussed the performance variations across different backbone LLMs in section 4.1.
>
> > **"Assess more than 30 examples of hypothesis validation failure modes"**
>
> Thank you for highlighting this aspect. To clarify, we initially analyzed 20 failed experiments to derive 10 distinct failure-mode categories. Subsequently, we expanded this analysis using a comprehensive set of **128** failure cases collected from benchmark runs across TargetVal-IFNG, TargetVal-IL2, and DiscoveryBench, as stated in Appendix D. Indeed, these include all the failure cases from one run of our experiments. Therefore, our analysis already provides extensive coverage, which we will clearly emphasize in the revision.
>
> > **"Run a synthetic experiment with known ground truth and simulated expression data"**
>
> We thank the reviewer for this insightful suggestion. We would like to clarify that  we indeed design the experiment setup for Type I error rate estimation to precisely address this concern. We simulated a null scenario by permuting rows of the dataset (e.g., shuffling gene names or expression values), thereby disrupting any real associations between variables. After permutation, all hypotheses become null—including those that may have been true positives in the original data—ensuring that our evaluation of the type-I error is faithful. Under this known null ground truth condition, POPPER consistently refrained from rejecting most null hypotheses, effectively controlling the Type I error rate. We will clarify this explicitly in the revised manuscript.

---

### Decision · Program_Chairs · 2025-05-01

**Decision:**

Accept (poster)

**Comment:**

This paper shows how to use an LLM to design experiments in a systematic way that aim to falsify a hypothesis expressed in natural language. The topic is important in current AI research.

Scores from reviewers are 3, 1, 4, 4. The review with low score makes valid points, but these are not directly about the technical content of the paper, which is a clear contribution.

Reviewers and the area chair and the senior area chair all agree that the original submission contains majorly exaggerated claims. The authors (a) must be realistic and accurate in their claims in the final version of the paper, and (b) must describe the limitations of the work clearly and prominently in the main paper.